# Positive Health Behavior Changes in Custodial Grandparents and Grandchildren Following an Intervention

**DOI:** 10.3390/ijerph19074027

**Published:** 2022-03-29

**Authors:** Christine A. Fruhauf, Angela Nancy Mendoza, Aimee L. Fox, Heather Greenwood-Junkermeier, Nathaniel R. Riggs, Loriena A. Yancura

**Affiliations:** 1Department of Human Development and Family Studies, Colorado State University, Fort Collins, CO 80523, USA; nathaniel.riggs@colostate.edu; 2College of Social Work, The Ohio State University, Columbus, OH 43210, USA; mendoza.794@osu.edu; 3Center on Aging, Kansas State University, Manhattan, KS 66506, USA; aimeefox@ksu.edu; 4Department of Family and Consumer Sciences, University of Hawai’i at Manoa, Honolulu, HI 96822, USA; heather8@hawaii.edu (H.G.-J.); loriena@hawaii.edu (L.A.Y.)

**Keywords:** grandfamilies, self-care practices, strengths-based approaches, health and wellness

## Abstract

Background: Custodial grandparents experience greater physical health declines and higher rates of depression than their same-age peers who do not provide care, and grandchildren in grandfamilies often have behavior problems. However, few researchers have explored the impact of self-care education on decreasing these negative outcomes. Our study examined how a self-care and life-skills intervention influenced health behavior change in a sample of grandparents and grandchildren. Methods: Data were collected during eleven focus groups (and two interviews) with 55 grandparents, and one focus group with five grandchildren, at 6 months after the 6-week intervention detailed in this paper. Grandparents ranged in age from 46 to 84 years old (*M* = 62.19, *SD* = 8.24). Participating grandchildren ranged in age from 9 to 12 years old. Focus group transcripts were coded for content related to grandparents’ and grandchildren’s positive behavior changes following the intervention. Findings: Grandparents reported taking more time for themselves, reducing negative self-talk, increasing healthy physical choices, and having better communication skills after participating in the intervention. Similarly, grandchildren reported increased confidence in making friends, making good decisions, and getting along with others. Findings suggest that a self-care and life skills program show promise for improving the health and wellness of grandfamilies.

## 1. Introduction

Grandparents raising grandchildren (i.e., with no parent present in the home) experience greater physical health declines and higher rates of depression than their age-related peers who are not primary caregivers to grandchildren [1,2]. The negative impact on custodial grandparents’ health and wellness may be further complicated by their experiences of strained resources, social stigma, and isolation from family and friends [1,3]. Skipped-generation homes may also struggle with the impact adverse childhood experiences have on the health and well-being of grandchildren, as they often experience physical, psychological, and behavioral challenges [4], and many grandchildren have had trauma exposure [5]. Further, researchers have discovered that anywhere from 30 to 50% of grandchildren raised solely by grandparents have both internalizing and externalizing symptoms [6,7]. Thus, grandfamilies (i.e., grandparents and grandchildren in homes with no parent present) are increasingly recognized as a vulnerable, multigenerational population that may benefit from preventative programming, interventions, and clinical approaches designed for grandfamilies [4].

Despite these challenges and adversities experienced by grandfamilies, grandparents and the grandchildren in their homes are resilient [8]. For example, Gómez [9] discovered that grandparent caregivers reported greater family resilience than other relative caregivers, pointing to the fact that grandparents’ previous parenting experiences might contribute to garnering family support. Bergeman and Wallace [10] outline individual (e.g., self-concept) and family and community support factors (e.g., social and community support and family environment) that may foster resilience processes, especially in later life. Among custodial grandparents, it is known that optimism, emotional stability, and confidence in parenting ability may help grandparents regain a sense of control to positively reframe caregiving challenges [8]. Social support and coping have also been found to be protective factors for grandparents in skipped-generation households [11]. For example, social support, including help or encouragement from friends and community members, can help promote self-care practices and utilization of services in grandparents [12]. In addition, fostering positive family communication may help grandparents and grandchildren communicate needs to each other and others, and maintain a sense of independence [13]. Positive attributes such as effective communication and emotion regulation can be taught with curricula designed to foster self-care and life skills.

Grandparents raising grandchildren have been of interest to scholars for the past forty years [1,14] and may benefit from programs improving specific well-being outcomes. Interventions designed for grandfamilies have predominately focused on targeting changes in grandparents; surprisingly few have designed interventions for both grandparents and grandchildren (i.e., intergenerational or systemic programs for grandfamilies [4]). This lack of attention to intergenerational programs for grandfamilies is not surprising, as interventions targeting grandparents and grandchildren may be difficult to design and implement. These difficulties are due to the common reasons (e.g., parental alcohol and drug use and abuse, abandonment, neglect, physical and emotional abuse towards grandchildren, parental death or divorce, and/or involved with child protective services) why the middle-generation (i.e., parent) may be absent from parenting, age and cohort differences between grandparents, parents, and grandchildren, and the diversity of grandfamily needs [1,4,15]. However, it is known that grandparent participants in training programs for skipped-generation families: improve their parenting skills [16]; increase levels of individual empowerment [17] and resourcefulness [18]; reduce psychological distress [19], implement self-care practices [20], and raise overall well-being [21].

As a result of the lack of intergenerational programs and the increased need for programs supporting grandparents and grandchildren, we designed, implemented, and evaluated an innovative intervention to improve grandfamilies’ overall health and wellness through self-care and life skills direct programming. The purpose of this article is to briefly discuss the theoretical underpinnings of the design of the GRANDcares intervention and report findings from separate focus groups with grandparents and grandchildren 6-months after program participation. (Please see [22] for a detailed description of the theoretical framework and the design of the GRANDcares intervention.)

### Theoretical Frameworks and Program Interventions

The guiding theories used to conceptualize the GRANDcares intervention and analyze the data presented in this paper are family resiliency theory (FRT [23]) and the strengths-based approach [24]. Walsh’s [23] FRT additionally informed our analyses, as it conceptualizes the whole family as a unit connected by dynamic processes [22]. Further, FRT supports scholars’ understanding of how families build protective factors and strengths when faced with traumas or crises. Thus, a strengths-based approach [24], grounded in social work practice and service delivery, closely aligns with FRT and assists us in understanding the assets, rather than deficits, that grandfamilies use to manage challenges [22]. Taken together, these two perspectives guided our design of the intervention and analysis for this paper as they are especially important when understanding the health and well-being of vulnerable families [25], such as grandfamilies.

In the grandparent portion of the GRANDcares intervention, we made small modifications (with permission) to the Powerful Tools for Caregivers (PTC) program, originally for parents raising children with special needs to render it appropriate for grandparents raising grandchildren (PTC-grandfamilies). The initial PTC program, designed for family caregivers, is highly effective and evidence-based, recognized by the Administration on Aging in 2012 as meeting the highest-level criteria for evidence-based programs. Emphasis is on self-care education, as its taught components are rooted in Bandura’s Social Cognitive Theory. It rests on the assumption that increases in individuals’ self-efficacy will increase the likelihood of the successful performance of tasks [26]. The PTC-grandfamilies program is implemented over 6-weeks. Over 2 h each week, participants develop eight self-care tools that: (1) reduce stress, (2) change negative self-talk, (3) communicate their needs, (4) recognize their emotions, (5) deal with difficult feelings, (6) make tough caregiving decisions, (7) set goals, and (8) problem solve.

The grandchildrens’ program (i.e., GRANDcares Youth Club) was created to complement the PTC-grandfamilies program, to support the needs of grandchildren aged 9–12 years. It is delivered at the same time as the grandparent program. Weekly topics and session components are parallel to the PTC-grandfamilies program but are age-appropriate for grandchildren and delivered in an experiential learning setting. The GRANDcares Youth Club is based on 4-H positive youth development theory [27], which assists youth in developing life skills (i.e., social conscience, personal values, caring, decision-making, and critical thinking) and relationships. Because we designed the GRANDcares intervention for grandparents and grandchildren, we encouraged grandfamilies participating in the PTC-grandfamilies and the GRANDcares Youth Club to discuss and practice the concepts they learned during each session over the week.

Participants in the PTC-grandfamilies and the GRANDcares Youth Club helped us to understand the impacts of these programs on self-care practices and health-related outcomes in grandparents, as well as on life skills and positive behavioral outcomes in grandchildren. The research questions guiding our focus group protocols and data analyses were: (1) What did grandparents and grandchildren learn about self-care and life skills that they continue to use 6 months after the GRANDcares intervention? (2) What individual and family-level protective factors were influenced by the grandfamilies’ participation in the GRANDcares intervention?

## 2. Materials and Methods

Focus groups were used to understand participants’ outcomes because: they are useful in gathering data from individuals who have had similar experiences [28]; are appropriate for assessing health-related interventions that take place in group settings [29]; have been used with both grandparents and grandchildren to assess effects of programming [30]. In addition, small focus groups with children ages seven or older are often used in the literature to understand their experiences [31,32]. Focus groups are also used in program evaluation because quantitative assessments may limit exploration to specific outcomes and do not allow for participants to elaborate on their experiences [29].

### 2.1. Procedures

In order to participate in the GRANDcares intervention, delivered in the states of Colorado and Hawaii in the United States, adults had to identify as the primary caregiver (i.e., grandparent or kin), with no parent present in the home, for one or more grandchild(ren) under the age of 18 years old. We did not place any exclusion criteria on the length of time that they had been raising grandchildren; however, we strongly encourage any grandparent that had been raising a grandchild for less than 6 months to consider delaying their enrollment. All grandchild participants were between the ages of 9 and 12 years old and were living in grandparent-headed homes. Grandparents and grandchildren who had participated in the GRANDcares intervention took part in one focus group at the 6-month follow-up. All focus groups were conducted by a trained research assistant and lasted approximately 45 min to 1 h. Responses from all groups were audio-recorded and transcribed verbatim. Identical questions and procedures were used in individual interviews with 2 grandparents who did not attend focus group sessions. Including both generations, 11 focus groups and 2 interviews with grandparents, and one focus group with grandchildren were conducted. Participants were prompted to address specific questions (see Table 1). Due to age and developmental differences among the generations, and differences in the programs (i.e., PTC is an established evidence-based program whereas the GRANDcares Youth Club is in its initial testing phase) different sets of questions were asked of grandparents and grandchildren.

### 2.2. Participants

A total of 149 grandparents, average age 62 years old (*SD* = 8.43, range: 39 to 83 years), raising 243 grandchildren under the age of 18-years-old (on average for 5.2 years, *SD* = 3.85), completed baseline assessments. Nearly three-quarters, 104 (70%), of the grandparents completed the intervention. At a 6-month follow-up, 55 grandparents (86% grandmothers), ranging in age from 46 to 84 years (*M* = 62.19, *SD* = 8.24) and who were raising a total of 83 grandchildren, participated in the focus groups/interviews. One focus group was held for grandchildren, 2 granddaughters and 3 grandsons ranging in age from 9 to 12 years old, participated in this focus group.

### 2.3. Data Analysis

Data analysis, using family resilience theory and strengths-based approaches as guiding theoretical frameworks, began after seven focus groups and one interview were completed and transcribed. Four authors read the transcripts line-by-line, taking notes in the margins about keywords and ideas relevant to our research questions [33]. With ongoing conversations and reflections among research team members, we put the data under intense scrutiny and formulated an initial coding scheme to represent the grandparents’ experiences after participating in the intervention. After all of the 11 focus groups and 2 interviews were completed and transcribed (please note: participant identities were not noted in the transcripts), two authors and a research assistant examined the additional focus group transcripts for alignment with the original coding scheme. New themes that more appropriately answered our research questions were identified and added to the scheme. All the grandparent data were then recoded with the new, more complete codes. Next, we used the constant comparison approach to condense the data into themes [34] and met to achieve 100% consensus on their meanings, each time refining the coding scheme to accurately represent the data. Finally, we examined data from grandchildren, and further assessed their alignment with the coding scheme arrived upon through the analysis of the grandparent transcripts, paying attention to emerging themes. After further scrutiny of the coding scheme, we discovered that data from the focus groups from both generations aligned with each other and represented changes in physical, mental/emotional, and social aspects of grandfamilies lives; thus, we added the biopsychosocial model of behavior [35] as a meaningful lens through which to interpret these changes. Thus, our final coding scheme was represented by three themes, each with their own code (see Table 2).

Our research approach ensured the trustworthiness and credibility of the analyses. For example, we engaged in peer examination [36], whereby initial findings from our analysis were shared at an international gerontology conference [37] and discussed with other grandfamily scholars. We also shared the findings of our procedures and coding scheme among GRANDcares’ site leaders and program assistants. Further, we reflected upon our research lens [38] and any potential bias during the analysis process. As a team, we collectively have over 80 years of experience in conducting research, as well as analyzing grandfamily needs and policies, in the fields of Human Development and Family Science, Gerontology, Prevention Science, and Social Work. Additionally, all co-authors believe that custodial grandparents and their grandchildren are resilient and seek to improve their health and wellness independently and with the assistance of educational programming. Finally, we used triangulation [36] of the literature, our conceptual model (see [22]), and data from focus groups, interviews, field notes, and grandparent brainstorming sessions, to strengthen the credibility of our research process.

## 3. Findings

Our analysis revealed that 6 months after participating in the GRANDcares intervention, grandfamilies experienced gains and improvements in health and wellness. Themes and codes depict grandparents’ and grandchildren’s health-related behavior changes, particularly in coping mechanisms, as they navigate challenges that arise from their family structures (i.e., the grandfamily). Further, in line with our guiding theories and conceptual model (see [22]), as well as post-hoc placement of focus group data within the biopsychosocial model, primary themes emerging from these data were: (1) grandfamilies’ engagement in physical and leisure activities, (2) grandfamilies’ ability to manage emotional stressors, and (3) grandfamilies’ establishment of social and community support systems. Even though we did not ask parallel questions during grandparent and grandchildren focus groups, we used the same coding scheme to analyze data and present the findings from both generations and found that both generations shared how the program helped them in their life. Furthermore, grandparents often discussed family-level outcomes that they believed benefited both generations. Therefore, when appropriate, we use ‘grandfamilies’ to describe such findings. In the following section, we discuss the grandparents’ data first, followed by information from the grandchildren.

### 3.1. Grandfamilies’ Engagement in Physical and Leisure Activities

The first theme, grandfamilies’ engagement in physical and leisure activities, was reflected in the grandparents’ descriptions of strategies learned as a result of the program and used to cope during stressful situations. Such activities included building on hobbies such as sewing, and engaging in leisure activities. One grandparent stated, “Mindfulness, you know and digging deeper into that, and with the GRANDcares class the journaling, of listening to music, you know that is calm and soothing”. In addition, grandparents reflected that they learned healthy coping strategies that they could use to manage their stress. For example, grandparents often discussed taking walks, hiking, doing yoga, and having plenty of rest. One grandparent went as far to share that her long-term action plan was to stop smoking. She stated that in the past 6 months, “I also quit smoking. I like, quit smoking”. Grandparents also discussed how other people (their grandchildren, partners, family members) supported them in staying physically healthy. For example, one grandparent shared:

That young boy is keeping me young. By playing football and come on let’s go. You know he wants me to go to the skate park with him and watch him and I do. And before I wouldn’t do things like that. … Now I feel better. More healthy in my mind. More because I’m stopping to give my time to him and going out and enjoying things like that.

### 3.2. Grandfamilies’ Ability to Manage Emotional Stressors

The second theme, the grandfamilies’ ability to manage emotional stressors, emerged as several grandparents discussed having hope and a positive outlook on life as they navigated situations perceived as being within their control. Many grandparents discussed how they let go of the need to control the situation, regardless of how it arose. Grandparents often discussed the importance of taking a ‘break’ or finding time to ‘breathe’ to calm themselves. This internal resource is illustrated in one grandparents’ statement, “I take a deep breath, go in my room and close the door and go … [breathes deeply] ‘okay, my room’s okay.’ Shut their door”. Moreover, when another grandparent stated, “The breathing technique. … Just take some deep breathes, leave the room, deep, deep breathes and it just calms me and makes me tackle the next hour, or whatever is going on”. Further, grandparents discussed an increase in self-efficacy. For example, a grandparent stated, “To realize and to depend on myself and not doubt myself as much. I was like ‘you know something you don’t need a college degree to raise a kid’” and another grandparent stated, “I would say, I’m probably stronger than I think and can be resourceful and make things work … strong even when I feel weak”. Further, another grandparent reflected on what had not been working with how she thought about her relationships, family, and control of the situation before the intervention. She stated:

I was so fixated on what am I doing wrong. What can I do better? Why aren’t things good. Why is my relationship failing, and when I let go of all that worry and just concentrated on what is and that I can’t change it, I don’t know it was like this thing happened. I was able to say, it’s okay to care for myself. It’s okay to let go. It’s okay to not make things be what you want them to be. Things can be okay without you forcing it.

This reflected the ways in which grandparents viewed relationships with their grandchildren and other family members in the context of social support and meaning, illustrating a pattern of wanting to control situations and, because of the class, becoming stronger in emotion-focused coping. For example, one grandparent stated, “Making time for him [grandson]. Making time for the wife and others you know what I mean? It’s not all about me”. It was evident by these grandparents’ reflections that the skills learned during the program were useful to their overall wellness in the face of relationships with other family members.

Further, as grandparents discussed their day-to-day experiences raising their grandchildren, they reflected they were better able to manage challenges as a result of their participation in the program. For example, one grandparent stated, “I used to think of life being like a rollercoaster with all its ups and downs and turns and twists and unexpected things, and now I think of it more as a road, and I do have some control”. Another coping strategy that grandparents discussed was learning to set boundaries. One grandparent illustrated this by saying, “Two things [I learned], one is don’t judge what you don’t know. And, the other one that I have been thinking of is ‘not my circus, not my monkeys”.

Additionally, grandparents often discussed that, as a result of the class, they are more confident in how they raised their grandchildren. For example, one grandparent stated, “It helped me build [my] own confidence level and [my] own self-esteem. We shared a lot of resources, which helped tremendously. I really loved this class because I know when I came in, I was like shocked still and crying”. Grandparents also discussed the changes they made to their lives as a result of this program and how these changes influenced their grandchildren. For example, one grandparent stated, “I’m more relaxed and that allows them (grandsons) to relax. … They’ve reflected that as well. They’re calmer”. Grandparents also discussed increased awareness, and one grandparent reflected, “[I’m] aware of like their [grandchildren’s] attitudes, their moods, [and] more mindful of what is more important to a healthy relationship with them and not so much yelling”.

Grandparents specifically mentioned implementing many of the tools they learned during class. For example, grandparents were taught how to use “I” statements and the importance of being assertive when communicating. Mastery of this skill was evident in the following grandparent statements, “Using “I” statements instead of you has improved our communication a lot” and “I find myself being a little more assertive, um that comes along with the self-care that I seem to be focusing on”. Furthermore, grandparents discussed how these helpful tools translated to better communication with their grandchildren. One grandparent stated, “Its working great cause now my grandson, I can talk to him and he looks at me and he respects what I say”. Another grandparent reflected how the changes she made resulted in her “(grandchildren) coming to me more and talking about … whatever’s bothering them”. Yet, grandparents still struggled with implementing these tools to control their emotions, while at the same time recognizing that it is okay to protect themselves from challenging situations and prevent further emotional challenges. For example, one grandparent stated:

I still struggled with the ‘I’ statement with my daughter, mother and granddaughters. She just … she can really piss me off [low voice; laughs]. And I, you know, it’s to the point where she … just her calling me. I don’t even … At times, I don’t just answer the phone. No one talks to her. And I feel, part of me feels bad, you know, this is my daughter. But, if I’m not careful, I know she just sucks the life out of me.

These communication skills served grandparents well in the context of managing themselves and connecting with their grandchildren. Many grandparents also reflected that it helped them in other challenging situations with other family members and schoolteachers. This was reflected in the following statement from a grandparent:

It has taught me a lot about communication and interaction, resources, and how to deal with members of the family and the school, and other people in general. It’s given me tools to help them [grandchildren] say things to people who might ask them uncomfortable questions.

Another grandparent mentioned how this helped improve connections among her granddaughter, her granddaughter’s teacher, and herself when she stated, “It has opened up even more than deeper communication between me and her teacher … and she was hugging her the next day. Using that technique was so, so good”.

In addition to grandparents, grandchildren also discussed how the 6-week program impacted their lives. For example, one grandchild stated that the youth club provided a way for her to “learn how to cooperate with each other” and that she thought it was “helpful for real life, teaches [us] how to use different tools and think about different things”. Another grandchild reflected that it supported, “Real life–used tools: [I] used to be really shy, [from the class, I] learned how to talk to other people, [and] now made a lot of friends”. Grandchildren’s reflection of how they took what they learned during the class and applied it to their lives illustrated the applicability of the skills they learned from the intervention.

Similar to what grandparents reported, grandchildren also mentioned gaining knowledge of how to manage themselves as well as access support from others. For example, one grandchild stated, “Set goals for ourselves. That was really fun. I still do the goals that I set in class”. Another grandchild reflected that they “trusted everything and everyone”. These grandchildren examples showcased the awareness and use of internal and external resources that they gained as a result of the program.

### 3.3. Grandfamilies’ Establishment of Social and Community Support Systems

The third theme, the grandfamilies’ establishment of social and community support systems, was illustrated when grandfamilies discussed their flexibility when dealing with challenges and their utilization of social and community resources when they needed them. For example, grandparents discussed using external resources through the local food banks, grandfamily coalitions/support groups, and counseling services. One grandparent stated, “that’s when I started really noticing, ‘oh, maybe you are a little depressed.’ So, I started in therapy”. and another grandparent needed help with cooking and stated, “using more the meal help services” was useful during stressful days. Additionally, many grandparents stated that finding other grandparents in similar situations through this program served as a support mechanism. This was illustrated by one grandparent:

The peer support is huge … I didn’t realize until I heard everybody’s story … and a couple of times I thought ‘wow! I’m part of this group now … and I think that’s huge you know. You’re out there and you feel like you’re alone … woe is me [laughs], I give up my life.

Moreover, another grandparent stated, “Know[ing] that others have the same issues. I couldn’t believe how similar [her—name of another participant] and I were with a lot of stuff”. Grandparents also shared how raising grandchildren often puts them in situations where they feel out of place, yet they feel connected to others since participating in this intervention. For example, one grandparent stated:

I liked just being with other adults who understood what it was like to be a grandparent because when I go to school to pick her up, everybody is in their 30 s with their leggings and being in good shape and here I come going like ‘I don’t fit into this picture.’

In addition, grandparents reported how they maintained contact with each other after the program ended. In particular, a few grandparents reflected on this during a focus group. One grandparent said, “you know I’ve connected with some of you outside a little bit and plan to do that more… I hope we keep meeting and do things outside in the moments we can steal”. Another grandparent in the same focus group replied, “[we] get together once a month or we try to. We have a date out there so if you can make it great….but that’s just been nice having, even if we get to see each other once every two months”. Then a third grandparent replied, “I think knowing that I am just part of this whole, …that I’m not alone. There’s a whole bunch of other grandmas out there exhausted trying to take care of their kids and it’s a very noble cause”. Thus, the peer-to-peer support and mentoring may have assisted in grandparents’ ability to further navigate challenging situations.

Grandparents whose grandchildren participated in the youth club reported that their grandchildren looked forward to the sessions and excitement in being around children who “are like me” or also part of a grandfamily. Grandchildren who participated in the focus group reflected that they enjoyed being around peers with similar family arrangements. For example, one grandchild stated, “you get to be yourself and get to have a lot of fun!” Further, reflecting the excitement for the program and connections made between grandchildren, another grandchild stated that, “The party was too short at the end [of the program]”.

## 4. Discussion

The findings of this study demonstrate that a grandfamily intervention grounded in a strengths-based approach fosters improvements in the self-care practices and life skills of both grandparent and grandchild. Above all, after having participated in the GRANDcares intervention, both generations discussed having learned useful coping strategies and skills that helped them to deal with stressors related to living in skipped-generation households. For example, grandparents discussed having a positive outlook and stronger connections to their grandchildren/other family members during the 6-month focus groups. Both generations reflected upon building problem-solving skills, being resourceful, and finding they had support from their peers (i.e., other participants), all protective factors for supporting family resiliency [23].

The findings from this study include individual and family-level changes. This is consistent with previous intervention research reporting that grandparents raising grandchildren can: improve their parenting practices [16,19]; increase resourcefulness and resiliency [18]; decrease psychological distress and depressive symptoms [19]; set solution-focused goals for the future [21]. The inclusion of grandchildren is unique to the current study as they are often neglected in grandfamily focused interventions [4].

The present study reveals that after participating in the GRANDcares intervention, both grandparents and grandchildren used and implemented skills learned during the program immediately after the program and through the 6-month follow-up. In response to our first research question, we discovered, through the reflections and discussion among grandparents, that changes in self-care and life skills took place among grandfamilies. For example, grandparents spoke about when they face a problem, they are better able to manage it, and they are empowered to take control. These grandparents seemed excited to use the skills they learned during the program to fix or change things, and they discussed how these changes positively influenced themselves and their grandchildren, including those grandchildren that did not participate in the GRANDcares Youth Club. Several grandparents gave examples of how taking breaks when under stress and using calming techniques during emotionally challenging situations not only helped their overall wellness but also provided their grandchildren with role-modeling of positive health behaviors. Grandparents further discussed seeing changes in their grandchildren’s behaviors, such as better listening skills and coping with school or peer-related stress. Grandparents reflected that they believed this was based on what they had learned during the PTC-grandfamilies program; thus, even when grandparents are the target audience for programming, such interventions may also reduce both internal and external challenges of their grandchildren [19]. As a result, perhaps what participants developed from this intervention is not only individual self-care skills, but they also develop family self-care skills.

Grandparents often described how the skills they learned from the intervention translated to their improved communication with their grandchildren, family and friends, their grandchildren’s teachers, and other healthcare and service providers. Trusting other people is not only important for individuals but may also translate to increased communication in families and communities [23], thus answering our second research question. As a result, the communication techniques delivered in the program may ultimately improve grandfamilies’ overall health and wellness. However, it is important for future research to further understand how interventions impact grandfamilies’ use of community-level resources. Similar to Mendoza and colleagues’ [11] research, our work also reflects that grandfamilies may benefit from interventions that help them to build support networks and learn proactive coping skills, as opposed to focusing only on mitigating stressors. Thus, improved intergenerational health outcomes in grandfamilies are supported by the processes of mutual support and problem solving [9]. Researchers may also be interested in further exploration of how interventions lead to grandparent empowerment as grandparents in this study noted their increased abilities to reduce stressful situations and to manage challenging discussions. This is not surprising given that the grandparents who participated in the larger study also showed improvement in caregiver self-efficacy [39] as they navigated their caregiving role to grandchildren.

Grandchildren who participated in the grandchild focus group reflected on what they learned during the GRANDcares Youth Club. They mentioned that they still use action plans to set individual goals and that they learned they could make friends and trust other people. Unlike their grandparents, grandchildren did not discuss strategies when dealing with stressful situations. Grandchildren only reflected on what they learned during the program, what they are still doing, and how they felt interacting with their peers during the program.

Many grandfamilies experience challenges that negatively impact their overall health and well-being, as well as threaten opportunities to develop family resiliency [9,22]. In our study, grandparents and grandchildren reported positive changes in physical, mental/emotional, and social aspects of their lives, in line with the biopsychosocial model [35] of illness. This model is rarely represented in the disciplines of Intergenerational Studies and Family Science. This model provides the additional insight that negative outcomes experienced by grandparents and grandchildren often involve physical and psychological factors, psychological traits and states, and social-environmental issues—as well as interactions and transactions among them. Perhaps future interventions designed to improve grandparent and grandchild health and well-being may benefit from multi-faceted approaches that purposefully address biological, psychological, and socio-environmental factors, and thus, implications for future research may consider using this approach.

We are cautious in our use of this lens, however, as this theoretical foundation is traditionally used in the context of understanding individuals in the midst of chronic illnesses [40] and not necessarily at-risk family populations. However, what we discovered is that grandparents and grandchildren discussed changes in all of these areas, perhaps contributing to the comprehensive considerations of grandfamily self-care. These healthy changes and engagement in physical and leisure-related activities [41] may link physical, psychological, and social changes and lead to increased meaning, thus benefitting grandparents and grandchildren. Future research may consider dyadic approaches to grandfamily interventions that focus on self-care as it pertains to the grandfamily as a unit as well as individual members.

### Study Limitations

This study has limitations that should be noted. First, not all grandparents who completed the intervention participated in a 6-month follow-up focus group. These grandparents might have responded differently to the focus group questions and thus, do not represent all participants. Second, grandchildren that participated in the focus group might have been influenced by what they observed their grandparents do or say as a result of their participation in the program. Although grandparents and grandchildren participated in separate focus groups, inherently, they could have influenced each other’s thoughts about the program. Further, perhaps the grandchildren’s framing of the life skills they learned was not because they cannot manage stressful situations, but a result of our grandchildren focus group protocol, which did not prompt the youths to discuss how they apply the skills they learn in everyday situations. Future research may want to use similar but developmentally appropriate protocol questions with both grandparents and grandchildren to further the understanding of how self-care and life skills change as a result of such intervention. Finally, although all cohorts were scheduled to conduct focus groups, due to COVID-19 and subsequent research restrictions during the final three months of the project, not all cohorts of grandparents and grandchildren were given the opportunity to participate in a focus group. New or different findings from the data might have emerged from their experiences with the pandemic and thus are not represented in the coding scheme. As a result, these findings are only representative of the grandfamilies that participated in the focus groups and do not represent all grandparents or grandchildren that completed the 6-week intervention.

## 5. Conclusions

Findings from this study describe grandparents’ and grandchildren’s implementation of skills following both generations’ participation in educational programming designed to improve their self-care practices and life skills. The findings advance our understanding of how family interventions using resilience and strengths-based approaches may enhance the health and wellness of vulnerable populations. In sum, these findings support calls to advance intergenerational programming supporting grandfamilies, with particular attention to meeting the needs of both grandparents, who are at risk of increased negative outcomes to their physical and mental health, and grandchildren, who are known to experience adverse childhood events and trauma. These multigenerational efforts should include skill-building techniques, group discussion and reflection, and respite opportunities for grandfamilies. With further program development and evaluation, such programming efforts may lead to additional insights into the most effective intergenerational interventions for grandfamilies.

## Figures and Tables

**Table 1 ijerph-19-04027-t001:** Focus Group Protocol Questions.

Grandparents	Grandchildren
What changes occurred in your life as a result of participating in the intervention? What did you learn about yourself that you did not know before, after participating in the program? What tools have you found most helpful in the self-care practices that you learned during the intervention? What changes have occurred in the way your family communicates or interacts because of what you learned during the program?	What did you like most/least about the program? How comfortable were you during the class? What activities did you enjoy the most/the least? What do you believe should be changed about the program? Would you recommend the program to your friends?

**Table 2 ijerph-19-04027-t002:** Qualitative Coding Scheme for Experiences of Grandparents and Grandchildren.

Themes	Codes
(1) Grandfamilies’ engagement in physical and leisure activities	Meditation and mindfulness; walks; hiking; yoga; sleep; activities w/grandchildren
(2) Grandfamilies’ ability to manage emotional stressors	Emotion regulation: problem solving; walking away/letting go/boundaries Positive outlook: hope; learning acceptance, reframing situations Communication: ‘I’ statements; active listening
(3) Grandfamilies’ establishment of social and community support systems	Social support, peer-to-peer mentoring; agency-related services; counseling services

## Data Availability

The data presented in this study may be made available on request from the corresponding author. The data are not publicly available due to confidentiality of participants. We are doing this because: (1) ethical, legal or privacy issues are present, (2) to ensure that data shared are in accordance with consent provided by participants on the use of confidential data.

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
