# Peer review of "Positive Health Behavior Changes in Custodial Grandparents and Grandchildren Following an Intervention"

_ijerph, 2022, doi:10.3390/ijerph19074027_

Round 1

Reviewer 1 Report

I like this original work-it is well written, relevant to grandfamilies, and is appropriately theoretically grounded. I have a few observations:

  1. on p. 2 it is unclear why the absence of the middle generation would make interventions for grandparents and grandchildren difficullt to implement-I would think otherwise.
  2. It might be helpful to set off the section on the PTC program and even describe its content more specifically
  3. I wonder about the independence of grandchildren's responses apart from their grandparents. If each was enrolled in the intervention, might not what grandchildren had to say in the focus group be in part influenced by the grandparents' sharing their own experiences with the program and/or discussing with the grandchild what he/she had learned, etc?
  4. I appreciated the care with which the coding scheme development was described, but given the sample size, I would like to see how frequently responses coded categorically actually emerged. Which was most common? Least common? This would apply to both the grandparents' and grandchildren's data.
  5. I got the impression that I would learn about grandparents' findings first and then about grandchildren's. Yet, the emphasis in both the results and discussion sections seemed to be on grandparents-more needs to be explicitly said about what grandchildren had to say-it needs its own section at the minimum.
  6. The coding scheme on page 5 seemed to be grandparent oriented. It may have been better to first develop a coding scheme for the grandchildren, given that their focus group queries were unique to the youth program, and THEN attempt to integrate them at some level with grandparents' responses. I found myself questioning the continued reference to grandfamilies which seemed to be quite a stretch given the diverse nature of the programs/questions asked in the focus groups between grandparents and grandchildren. They seem to be quite different (p. 4)-one was more abstract/self-oriented while the other seemed more concrete/class oriented-are they really mergeable into 1 coding scheme?
  7. On p. 12 ,the authors need to say "this" theoretical foundation rather than "the" theoretical foundation.
  8. The authors clearly have data to support their guesses about the selectivity of those grandparents and grandchildren who participated in the focus groups vs. those who participated in the programs per se. This should be presented.

Reviewer 2 Report

Dear Authors!

The topic is very relevant, and the endeavour to address family resilience of grandfamilies is valuable. There are a few issues I would ask to modify in order to confer more clarity to the paper.

Wishing the authors good luck for this valuable activity,

best, reviewer

  1. Please state more clearly that grandfamilies are families where only grandparents raise the children, and parents are missing. Sometimes this is clear, but in the first sentence in the introduction it is statet: Grandparents raising grandchildren..., which might imply that they only contribute, i. e. assist parents. It should be added that they raise grandchildren exclusively! The term custodial grandparents is very adequate in the abstract and may be applied at some paces in the body text, too.
  2. The description of the interventions, starting from page 3 up: The PCT grandfamilies program.... rather belongs to the Materials and Methods section, please move there. Only the research questions at the end of this section should be left here.
  3. Participants section - where did the intervention take place? We do not find out even the country.
  4. Most importantly, please use a different text style for longer citations which last over one short paragraph, and put these texts into quotation marks: That young boy is keeping me young.... I was so fixated on what I am doing wrong.... The peer support is huge.... I liked just being with other adults..... O still struggled ..... It has taught me a lot about communication..... All these parts should be edited so that it is obvious that they are parts of the focus groups texts!
  5. Please avoid stating: One grandparent summarized this perfectly, by saying:.... Such judgments (perfectly) should be avoided.
  6. The Limitations should be a separate short chapter and not just added to the end of the paper. The part of the Discussion section: Perhaps grandchildren"s framing..... to understand their self-care and life skills changes should be also moved to the Limitations section.

Reviewer 3 Report

This research has an original objective and a meaningful content. I congratulate the authors for the work done. I am grateful with the editors for the possibility of revising this manuscript. 

Introduction

The introduction is clear and well worked.

Materials and Methods

Study design

The study design is appropriate and well described.I would like to indicate what were the inclusion and exclusion criteria of the participants since it is not very clear how their selection is established.

Discussion

Discussion is well oriented. 

Author Response

Thank you for your positive review of our work and for pointing out we did not include inclusion and exclusion criteria. We included this information on page 4, section 2.1.

Round 2

Reviewer 1 Report

no substantive changes are required-the authors' changes have improved the paper